# Recent Analysis of Forged Request Headers Constituted by HTTP DDoS

**DOI:** 10.3390/s20143820

**Published:** 2020-07-08

**Authors:** Abdul Ghafar Jaafar, Saiful Adli Ismail, Mohd Shahidan Abdullah, Nazri Kama, Azri Azmi, Othman Mohd Yusop

**Affiliations:** Razak Faculty of Technology and Informatics, Universiti Teknologi Malaysia (UTM), Kuala Lumpur 54100, Malaysia; saifuladli@utm.my (S.A.I.); mshahidan@utm.my (M.S.A.); mdnazri@utm.my (N.K.); azriazmi@utm.my (A.A.); othmanyusop@utm.my (O.M.Y.)

**Keywords:** DDoS, HTTP DDoS, GET Headers

## Abstract

Application Layer Distributed Denial of Service (DDoS) attacks are very challenging to detect. The shortfall at the application layer allows formation of HTTP DDoS as the request headers are not compulsory to be attached in an HTTP request. Furthermore, the header is editable, thus providing an attacker with the advantage to execute HTTP DDoS as it contains almost similar request header that can emulate a genuine client request. To the best of the authors’ knowledge, there are no recent studies that provide forged request headers pattern with the execution of the current HTTP DDoS attack scripts. Besides that, the current dataset for HTTP DDoS is not publicly available which leads to complexity for researchers to disclose false headers, causing them to rely on old dataset rather than more current attack patterns. Hence, this study conducted an analysis to disclose forged request headers patterns created by HTTP DDoS. The results of this study successfully disclose eight forged request headers patterns constituted by HTTP DDoS. The analysis was executed by using actual machines and eight real attack scripts which are capable of overwhelming a web server in a minimal duration. The request headers patterns were explained supported by a critical analysis to provide the outcome of this paper.

## 1. Introduction

Due to the increase of HTTP DDoS attacks against web servers, the need for a dataset for the attack is pertinent. However, recent datasets for HTTP DDoS attacks are not publicly available due to several reasons as noted by many studies. Behal and Kumar [1] conducted a study on DDoS dataset utilized by prior researchers and explained that publicly available dataset had its limitations as most of the datasets were captured from the network layer thus obscuring information that are available at the application layer. The lack of current dataset disclosing recent false GET request headers produced by HTTP DDoS attacks provides little knowledge on the pattern types and ways they are executed. HTTP DDoS utilizes request headers as a dummy to conceal malicious activity and send substantial amount of HTTP requests to overwhelm a web server. Aside from that, prior studies stated minimal request headers pattern constituted by HTTP DDoS and unable to deliver detailed explanation. The experiment which specifically points to forged request headers generated by HTTP DDoS with usage of a recent attack script is still not presented in research community. Hence, a comprehensive experiment is required to deal with real devices and attack scripts to disclose the current pattern of forged request headers adopted by attackers to make the attack appear authentic.

The dependency on publicly available dataset when conducting experiments for HTTP DDoS should be avoided as the datasets contain old attack patterns which contradict with the current attack patterns produced by current attack scripts. Jazi, et al. [2] explained that the lack of dataset poses a major difficulty to conduct an experiment. Singh, et al. [3] lamented that many studies employed old datasets and used university or organization’s web logs as the benchmark to evaluate their work and noted that comparing their work with an old dataset is meaningless. Thus, generating dataset that is close to the actual environment using network topologies is required [4]. Usage of current DDoS dataset permit researchers to acquire the latest atmosphere the forged request header employed by cyber intruders to make the attack traffic look authentic. The obsolete dataset like (1) KDD Cup Dataset 1999, (2) CAIDA DDoS Attack Dataset 2007, (3) Environmental Protection Agency (EPA) HTTP dataset 1995, (4) DARPA DDoS attack dataset 2009, (5) Clarknet 1995, (6) NASA 1995 has been noted by prior studies [1,5,6].

This paper reveals the forged request headers constituted by HTTP DDoS starting with Section 5 which discusses HTTP manipulation while Section 6 discloses invalid request headers utilized by attackers when launching HTTP DDoS attacks. Both sections help future researchers to determine the appropriate detection that can be applied when detecting malicious request produced by HTTP DDoS. Detection is the first defense to recognize malicious traffic unlike other methods such as prevention, mitigation and monitoring. Munivara Prasad, et al. [7] noted that detection is the first step to be executed followed by mitigation. This is because clean and attack traffics can be differentiated at the detection phase. According to Zargar, et al. [8], the deployment of the detection stage can be done either at the source, intermediate network, destination or a combination of them. HTTP DDoS generates massive HTTP requests to overwhelm a web server. Thus, selecting the most appropriate detection technique is very crucial to avoid higher rate of false positive and false negative. To ensure that a higher rate of true positive and true negative can be achieved, three detection methods broadly known as signature-based, anomaly-based, and hybrid must be deployed with precaution. This is to ensure that any detection flows that have been designed will be compatible with any of the intrusion prevention systems [9]. Thorough explanation pertaining to the different types of detection methods are presented in the following sub-sections.

### 1.1. Signature-Based Detection

Osanaiye, et al. [10] elaborated that a set of policies were applied to inspect incoming traffics and to utilize the knowledge database to store known attack patterns. The database was constructed by collecting all related attack patterns [11]. Traffics receive were compared with knowledge database which therefore required the database to be frequently updated to ensure the attack traffic can be detected [10,11,12]. However, adoption of absolute database leads to a misclassification of false positive and false negative due to a high possibility of occurrence. Cheng, et al. [13] noted that this approach requires information from the available dataset to capture the behavior and information about various attacks. The signature-based detection was built based on the network behavior and there are also various terms that have been used to describe it including misuse, knowledge-based, rule-based and pattern-based detection [13,14]. The signature-based detection depends on a knowledge-based database. A set of policies is required to detect malicious traffics; hence, usage of appropriate policy is highly recommended to be applied in order to recognize any attacks. An inaccurate policy will increase the occurrence of DDoS traffic to reach a web server. Aside from that, the signature-database requires frequent updates which will increase the storage capacity. As explained by Cheng, et al. [13], this approach uses available dataset to gain information about the attack. However, not all dataset is shared publicly due to security reasons. Due to this, collaboration with other parties is highly required by sharing the attack pattern to ensure that the database is constantly updated. Usage of security tools by performing experiments can also predict any incoming attack, which will increase the maturity of the knowledge database.

### 1.2. Anomaly-Based Detection

Myint Oo, et al. [14] noted that this detection type utilized a machine learning concept and is also known as an outlier and performance-based detection [13]. Cheng, et al. [13] further elaborate that anomaly-based detection is capable of detecting new and unknown attack types. Osanaiye, et al. [10] explained that anomaly-based detection utilizes a traffic’s profile of typical behavior to detect malicious behavior [10,12]. The traffic profile must operate for several days to acquire normal behavior access pattern. This technique utilizes two approaches noted as training and detection phases. The training phase relied on the input to obtain regular access behavior. The type of detection consisted of several categories such as supervised, semi supervised and unsupervised. In supervised learning, all data are labeled to presume the results while in a semi supervised learning, the data are not entirely labeled, and in unsupervised data, all data are unlabeled as it will learn based on the input data. Anomaly-based detection requires several days to thoroughly learn about normal user access behavior. Adopting this approach is time-consuming as all traffic behaviors must be stored in the traffic profile. An inaccurate classification will occur if the training profile is used before obtaining the entire behavior. The traffic profile also requires constant updates, especially for new access pattern, which is also equal to the signature-based detection as failure to update will result in malicious traffic to reach a web server.

### 1.3. Hybrid-Based Detection

This detection method merged signature-based and anomaly-based techniques to form a hybrid detection system [10,13,15]. Hybrid-based detection utilizes a feature from both techniques to obtain a higher detection rate [10]. Cheng, et al. [13] highlight that hybrid detection can help analyst to detect both normal and malicious system behavior to increase the capabilities for monitoring the detection system.

## 2. Contribution of the Paper

This paper aims at giving the reader thorough insights into the forged request header patterns generated by HTTP DDoS. To the best of our knowledge, there are no current studies conducted that look into HTTP DDoS with the adoption of recent attack scripts to disclose forged request headers adopted by the attack. The request headers pattern for HTTP DDoS categorized as slow rate HTTP DDoS has been disclosed by several studies [16,17,18]. However, the forged request headers for HTTP DDoS categorized as session flooding and request flooding are not excessively revealed. Hence this paper is an extension of surface knowledge provided by prior studies and contributes as follows:Eight forged request header patterns produced by HTTP DDoS have been revealed through extensive analysis by adopting actual HTTP DDoS attack scripts. Critical analysis has been made to acquire similarities and differences and a summary of the request header patterns.The HTTP DDoS architecture has been introduced based on the usage of eight actual attack scripts. The attack scripts have been adapted to have different attack strategies which are capable of executing HTTP DDoS through Direct Attack, Proxy, and Spoof IP address.A taxonomy of HTTP DDoS attack patterns has been created by incorporating forged request headers noted by prior studies with false request headers unveiled in this study.A recent attack script to execute HTTP DDoS has been assigned to the existing attack strategy suggested by prior studies.

## 3. DDoS Attack at the Application Layer

The protocol known as Transmission Control Protocol (TCP) and HTTP utilized by attackers to gain server resources lead to the inability of a web server to handle client requests [19]. Hoque, et al. [20] mentioned that the difficulties to detect DDoS at the application layer is due to three factors. The first factor is due to obscurity as an HTTP protocol uses TCP and UDP connections to run its operation thus leading to complications to differentiate between legitimate and illegitimate traffic. The second factor is efficiency as DDoS attack at the application layer only requires fewer connections to initiate an attack. The third and final factor is lethality as the attack has the capability to overwhelm a web server instantaneously resulting in service breakdown regardless of the type of hardware and its performance. A DDoS attack which occurs at this layer consumes resources, including CPU and memory [21,22].

## 4. DDoS Attack Strategy at Application Layer

The existence of DDoS attack strategy executed at the application layer is due to the architecture of a web server. For instance, a web server that handles a website allows users to access a web page without going through the main page. This approach provides convenience for users to search for information without accessing the first page of the website. As a result, attackers are able to launch attacks at different pages of the website. The attack strategy, as described above, indicates that the structure of the website allows the attacker to create several attack strategies to overwhelm a web server through a full HTTP request.

Conversely, for a web-based application that is commonly categorized as an online system, users will not be allowed to access the subpage without passing through the login page, which deny the attacker to execute DDoS at application a different page as explained above. The login page’s existence will deny attackers from targeting specific pages as the login page has blocked it; however, having the login page is only suitable for an online system and is inappropriate to be adopted on a website. A website is utilized by many entities to promote and provide info about a specific operation. Execution of DDoS attacks at the application layer employ several attack strategies. This is to ensure that the attack can bypass security devices, and the targeted web server is collapsed and unable to serve a client’s request. DDoS attacks are executed at the network layer, which can be easily detected [23]. Due to this, the attacker opts to execute the attack at the application layer to make a web server turn to offline mode therefore incapable of responding to a client’s request [3]. The attack strategy to execute DDoS attack at the application layer has been revealed by the same authors [3,6] in different studies as elaborate as follows:**Server Load:** Attacker uses botnet to continuously send malicious request against a web server aggressively, which will lead to the server to drop legitimate request as the resources of the web servers are running out.**Increasing:** Attacker uses low request value to initiate an attack and slowly increasing the value. This behavior of attack is difficult to detect as malicious HTTP traffic is not sent aggressively to a victim’s server during the occurrence of the attack.**Constant:** To perform this attack strategy, cyber intruders must specify a specific number of request rate to be sent to the victim’s HTTP web server. The request number is referred to as constant, which will be the same as when the botnet sends a malicious request to a web server. Commonly, the request rate is between 100, 200 and 300 per second.**Single Web Pages Attack:** Attacker uses any single web page that belongs to a website. The botnet is commanded by the attacker, which will continuously send malicious HTTP requests to web server.**Main Page Attack:** Cyber intruders specifically focus on the main page of the websites to deny legitimate users from getting any access. To generate an attack, traffic botnet will be used to repeatedly send a malicious request to a web server. The impact of this attack only occurs on the main page of the website, while the subpages of the website are not affected.**Dominant Page Attack**: This category refers to a web page with a greater interest for authentic users to access. The attacker then focuses on that particular page to embark HTTP DDoS attack to prevent a legitimate user from browsing the contents of the web pages. This attack only gives an impact on web pages that have a greater interest for users to browse.**Multiple Page Attack:** A cyber attacker will initiate the attack at multiple web pages from a website. The purpose of this technique is to avoid detection as the malicious HTTP request will imitate the human access pattern. For instance, humans will open more than one web page to find information while surfing websites. When the attack occurs, more than one of the existing web pages will not be accessible as the attacker is interested in targeting multiple web pages.**Reply Flood Attack:** The botnet command by the attacker sends an HTTP traffic at an inflated rate to gain a resource of the web server to prevent web server from surfing legitimate HTTP request. The attacks work by gaining human access patterns to prohibit the detection system from blocking the malicious request.**Random Attack:** All web pages that belong to a website are not accessible regardless of its category.**Rare Change Page Attack:** The common structure of a web system will group a page into a specific group to make the page content more structured and user friendly. Since the arrangement of the web page is grouped, an attacker may compromise the group page by commanding botnet to the web page. Hence, the group web page will be the most targeted pages. Consequently, this attack will prevent a user from opening a web page that belongs to a specific group.**Frequent Change Attack:** An attacker performs an attack on a web page that belongs to different categories. This attack will rotate and send a malicious request to distinct web page categories. This attack only affects specific categories of the website. When the attack occurs, other web pages can still be accessed as usual.**Hot Pages Attack:** Each web-based system will have frequent open pages. Hence, this situation will give an opportunity for cyber intruders to initiate the attack on the most visited pages to prevent legitimate users from accessing as the main objective for a DDoS attack is to avoid users from opening the pages.**Web Proxy Attack:** Attackers use a proxy server as a representative to generate attack traffic. The use of a proxy server to generate attack traffic will cause difficulties in detecting the source of the attack. Multiple proxy servers would be used to generate plenty of HTTP requests to overwhelm the web server.

## 5. HTTP Manipulation

Currently, HTTP is the protocol which is widely utilized by many web servers. Normal browsing involves two methods known as request and response to allow a user to browse through the web server content. Both methods contain headers which can reveal web applications and their infrastructure [24]. The request comprises headers such as “Accept”, “Accept-Encoding”, “Accept-Language”, “Authorization”, “Connection”, “Content-Length”, and “Content-Type” [25]. Tyson, et al. [26] noted that HTTP become vulnerable to be manipulated by intermediate parties once it communicates across a network.

Another request header which commonly existed in a request is known as “referrer” which refers to the previous website accessed by a user to access the current web site. The referrer is one of the headers adopted by the attacker to generate HTTP DDoS, thus making the request look authentic. Mansoori, et al. [27] noted that referrer was the standard header being exploited, which can be modified and spoofed by firewalls and proxies. A content type is another header that appears in a request which indicates the extension of the HTTP Uniform Resource Identifier (URI) header field [28]. However, the content-type is also vulnerable to be tampered as it appears inconsistently in the normal web browsing and can be declared manually [28]. Manually declaring content-type indicates that the header can be spoofed, which provides advantages to the attacker to emulate this header that resembles an authentic request. A user-agent is one of the compulsory headers that appears in a request header as noted by Liu, et al. [29]. A request will be considered as abnormal if the user-agent is not present. Usage of the user-agent is vital as it is used to determine the platform adopted by the user including the type of web browser, version, operating system, etc. More importantly, the information included in the user-agent is utilized to select the appropriate web content view which is either mobile or desktop view. Without the information provided by the user-agent, smartphone users may face difficulties in viewing the web content compared to the desktop view as there are differences between the two types of views. However, according to La, et al. [30], the user-agent can be modified by the attacker to perform the attack such as through Simple Query Language (SQL) injection, cross-site scripting and DoS.

Although attackers take advantage of the drawbacks that exist in the request header to execute an attack, security researchers can adopt the headers to design a defense system to recognize malicious traffics that used request headers to make the request looks authentic. Niu, et al. [31] propose a malware detection by using request headers such as “URI”, “Host”, “User-Agent”, “Request-Method”, “Request-Version”, “Accept”, “Accept-Encoding”, “Connection”, “Content-type”, “Cache-Control”, and “Content-length”. They explained that a malware requires to establish connection to Command and Control (C&C) server which has shorter request message. Saleh and Abdul Manaf [32] combined non-HTTP features with “user-agent”, “accept” and “host” while Yadav and Selvakumar [33] also adopted non-HTTP features to be combined with request headers known as “User-Agent” and “Referrer” in providing solutions to detect HTTP DDoS attacks. Malicious request traffic has a specific pattern which can be recognized through detection. Hence, Aceto and Pescape [34] proposed a sensor device that is capable of recognizing HTTP manipulation. The sensor utilizes HTTP response code to send message to a requestor once an HTTP has been tampered. Niakanlahiji, et al. [24], on the other hand, adopted the response header in recognizing phishing websites.

Adoption of attributes existed at the application layer such as request and response headers provided significant impacts in detecting of malicious traffics. This has been proven by prior studies, as noted above, that use the application layer headers capable of detecting an attack. HTTP DDoS is complex to detect due to its similarities with a legitimate request. Therefore, to recognize this attack, the current forged headers produced by the attack require to reveal excessively to acquire current header pattern adopted by the attacker in manipulating request headers. Section 5 discloses extensively the request headers constituted by HTTP DDoS.

## 6. HTTP DDoS Request Headers

DDoS at the application layer adopted request headers to make an HTTP request looks authentic. Consequently, the attack is complex to be recognized. There are several past studies that reveal the request headers employed by DDoS attacks and are further explained in the following sections.

### 6.1. HTTP DDoS Request Headers (Session Flooding and Request Flooding)

The request headers adopted by HTTP DDoS are categorized as session flooding and request flooding which are capable of mimicking a genuine user request [35,36,37,38]. Aside from that, the attack has the same syntax and is delivered via multiple HTTP requests in different HTTP formats [35]. Sreeram and Vuppala [37] contend that an excessive search request and login is one of the patterns of HTTP DDoS attacks while Yadav and Selvakumar [33] noted that HTTP DDoS attacks contain diverse URL and comprise of headers known as user-agent and referrer. The request headers included in HTTP DDoS comprised of an equivalent user-agent including legitimate, incorrect URL and a repeatable request against the equal URL [39].

### 6.2. HTTP DDoS Request Headers (Slow Rate)

The request header patterns for HTTP DDoS in the slow rate category contradicted with request headers employed by session flooding and request flooding attacks as only a single Carriage Return Line Feed (CRLF) exists at the last request headers [16,17,18]. The genuine request headers are supposed to have two CRLF. The HTTP request transaction contains a request header that will end with a CRLF which refers to the line break of each request header. Yevsieieva and Helalat [17] noted that the appearance of \r\n\r\n shows the headers are complete and indicates the beginning of the body message. The explanation provided in the studies was in line with RFC 2616 guidelines, in which CR refers to \r and LF refers to \n which make the genuine HTTP request to contain \r\n\r\n in the last request headers, indicating the line break of each request headers. Figure 1 illustrates the genuine request headers while Figure 2 indicates fake request headers.

The authentic CRLF can also be referred as American Standard Code for Information Interchange (ASCII) code that has a value of 0d 0a 0d 0a which represents two CRLF (CRLF, CRLF). However, during the occurrence of HTTP DDoS attack in the slow rate category, only 0d 0a (CRLF) existed [18]. Figure 3 demonstrates the legitimate request headers in ASCII code while Figure 4 indicates the ASCII code for the HTTP DDoS attack.

## 7. Analysis of Forged Request Headers

HTTP DDoS comprises many attack strategies as explained in prior studies in Section 4. However, none of the explanations reveal the forged request headers excessively adopted by the attack. HTTP DDoS delivers similar request headers as genuine requests to make the requests look real. Therefore, an extensive analysis is required in order to provide more accurate ingredients when dealing with the attack. The pattern must be understood with the adoption of a current attack script so that the outcome will benefit future studies to understand, propose and enhance the existing HTTP DDoS solution. Section 6.1 disclosed the invalid request headers utilized by HTTP DDoS categories including session flooding and request flooding while Section 6.2 reveals the false request headers utilized by HTTP DDoS categorized as a slow rate. The review of prior work which mentioned about false request headers adopted by HTTP DDoS in Section 6.1 delivers strong indication to mention that the forged request headers produced by HTTP DDoS categories as flooding require extension as past studies provide insufficient information and explanation pertaining to forget request headers adopted by the attack.

## 8. Materials and Methods

This study adopted a physical hardware when analyzing and disclosing the adoption of request headers in HTTP DDoS. Three attack scripts were executed in the internal network and five attack scripts were executed in an external network to obtain similarities and differences of request headers executed in different network topologies. The attack scripts launched in the public network had capabilities to generate HTTP DDoS traffic through Proxy and Spoof IP address. The attack duration for an internal HTTP DDoS attack was set to 10 min as it is sufficient to acquire the false request headers and investigate the attack pattern due to the high speed of the attack that are capable of generating substantial HTTP requests within a second. Aside from that, for HTTP DDoS through external network, the time period must be lower and was set to 5 min to reduce the impact as the traffic is blocked by ISP and host provider. Behal and Kumar [1] elaborated that evaluation of DDoS in the actual environment reduces the network performance due to plenty of traffic sent by the attack. The required hardware and software in this analysis are shown in Table 1 while Figure 5 illustrates the network architecture.

The attack script’s execution was conducted separately to ensure the forged request header produced by each attack script can be identified quickly. Execution of the attack in a distributed manner leads to complexity in differentiating malicious and legitimate requests. This approach allows this study to properly elaborate invalid request headers adopted by HTTP DDoS in making the request to appear authentic. Furthermore, HTTP DDoS only requires minimal botnet with the help of an efficient attack script executed in a single machine is sufficient to overloaded a target [21,40].

Ghafar A. Jaafar, et al. [41] explained future studies should utilize a self-generated dataset with the adoption of the attack script which is publicly available. This study utilizes eight attack scripts as indicated in Table 2 to conduct the analysis. Eight attack scripts were deemed as sufficient to conclude forged request headers generated by HTTP DDoS as there are similarities between the scripts in forging the request headers. The critical analysis conducted in Section 8 extensively elaborate similarities and differences between each script. The usage of these attack scripts are publicly available, and have been mentioned and employed broadly by many earlier studies [2,35,36,39,40,42,43,44]. Although DDoS as services available as noted by Hameed and Ali [45], the usage of attack scripts to be executed in local area network is preferable as DDoS as services require transmission of attack traffic across the network which increases the possibility of the attack traffic to be detected and blocked by many intermediate networks managed by the internet service provider (ISP). The results of the analysis will complement the existing HTTP DDoS request header patterns discovered by prior studies and are thoroughly explained in Section 6.1 and Section 6.2

## 9. Execute Attack Scripts

This section highlights the request headers adopted by HTTP DDoS attacks. The request headers that were utilized such as user-agent, referral, query and other request headers that were involved during the genuine request transaction that existed in the HTTP DDoS are disclosed.

### 9.1. Analysis 1–Attack.py

An attack script known as Attack.py contains 680 false HTTP requests which is nearly similar with genuine request headers. An initial analysis found that the attack scripts adopted random user-agents. The attack script was executed by using the python Attack.py -t 10000 -c 20 http://lab.com.my command. Various user-agents were employed which caused complexity in distinguishing the authenticity of the HTTP request. Further analysis found that the attack randomly constitutes the same request query to a web server. Huge request queries were created with a combination of capital letters to overwhelm a web server. Aside from that, the attack script also comprised of irrelevant HTTP referrals which are not relevant to be the web server’s URL source. A valid HTTP referral typically originates from the right sources such as from Google search engine or other relevant sites that are linked to the web server’s content. Table 3 illustrates the false user-agent and request query followed by Figure 6 and Figure 7 that indicate the false referrer generated by the attack script.

### 9.2. Analysis 2–Chihulk.py

An attack script known as Chihulk.py was executed by using the Python Chihulk.py http://lab.com.my command. This attack script generated 88769 lines of forged HTTP requests which contain additional user agents to create forged user agents such as mobile devices, web crawler, PlayStation, and open source operating. The existence of user agents known as the web crawlers such as bingbot from MSN and Googlebot from Google indicates that the web server‘s contents have been indexed by the search engine which will increase the trust that the source of the request originated from a genuine requestor. Further analysis revealed that the attack script generated a request query with a combination of numbers and segregated by a backslash besides being repeatedly sent and continuously generated. Table 4 indicates the forged user agent generated by the attack script and the request query.

Further explorations successfully reveal that the request query and the HTTP referral were adopted to generate false HTTP requests. The request query appears to be genuine as the accurate format was generated. However, the request query’s value is suspicious due to its disconnectedness to the web server’s content. Although the value of HTTP referral is the correct URL, it is irrelevant to be accessible from the location. Table 5 shows the HTTP referral attack with the request query while Table 6 presents the comparison between authentic and illegitimate HTTP referrals that were attached with the request query.

### 9.3. Analysis 3–High Orbit Ion Cannon (HOIC)

This attack script generated 97,332 false HTTP requests. This attack script focused on generating plenty HTTP requests without completely replicating genuine request headers. These attack scripts contain minimal false request headers such as Accept, Accept-Language and Host which are contradictory to authentic request headers which have more headers. A comparison with legitimate HTTP requests indicates that a complete header was supplied, which contradicts with false request headers generated by the attack script which only had minimal headers. Table 7 indicates the comparison between the complete and incomplete request headers.

### 9.4. Analysis 4–Golden Eye.py

Another attack script referred as the Golden Eye.py is required to run by using the python GoldenEye.py http://42.1.63.189. The attack scripts generated 21257 forged HTTP requests that were identical to legitimate request headers. The attack scripts constituted almost similar request headers delivered by genuine HTTP request such as user-agent, accept-encoding, connection and referrer. However, the value of HTTP referral cannot be regarded as the URL source as they are unrelated to the web server content and appears to be inconsistent. Figure 8 indicates the irrelevant referral while Figure 9 shows the missing referrer that generated from the same source of HTTP request.

To disclose further how the attack works, an extensive analysis is needed, which successfully reveals that the attack script creates a false longer request query with a combination of small letters, capital letters, symbols and numbers. There are important things that need to be highlighted, for example, the request query constituted by humans began with a question mark (?) and is meaningful compared to the request query generated by the server to fetch items started with the symbol (/). Figure 10 demonstrates the request query generated by the attack script followed by Figure 11 which indicates the request query created by the web server to fetch items.

### 9.5. Analysis 5–BlackHorizon.py

The BlackHorizon.py requires the python BlackHorizon.py http://42.1.63.189 command to execute; and it generated 23,974 invalid HTTP requests which are likely to appear as legitimate request. Initial inquiries indicate that all the required request headers were supplied. However, a detailed analysis shows that the attack script comprises afalse request query which contains capital letters, small letters, numbers, and symbols. Further analysis revealed that the existence of an incorrect header in HTTP requests such as Keep-Alive has been found. The Keep-Alive is the value of the header in HTTP responses. However, it has been missused by attackers to be attached to the header. Figure 12 illustrates the request header supplied by the attack script while Table 8 indicates a comparison of valid and invalid KeepAlive values which involve in HTTP request.

### 9.6. Analysis 6–Wreckuests.py

The Wreckuests.py attack script generated the HTTP DDoS traffic through spoof IP address and required the python3 Wreckuests.py -v http://42.1.63.189189 command to execute. The attack script generated 9397 false HTTP requests that were sent through spoof IP address. Initial analysis disclosed that the attack script generated meaningful request query which can be understood by humans even when the query was unrelated to the web server’s content. Further analysis revealed that the source of the spoof IP address and software known as ZenMap were utilized. Three IP addresses were opted randomly, and the outcomes indicate that IP address 62.210.15.199 belongs to a web server and utilized port 22, 80, 443 and 3306 to operate. Next, IP address 92.222.74.221 was selected and the result showed that the IP address was assigned to a web server that utilized port 22 and 80 to operate. Finally, IP address 103.234.254.10 was analyzed and the outcome indicated that the IP address is owned by a router and adopted port 53 and 1723 to operate. The usage of this attack script indicates that the source of HTTP DDoS attack can also originate from IP spoofing such as the server and router. Figure 13, Figure 14, Figure 15 and Figure 16 indicate the request headers that were made by the three IP addresses discussed earlier while Figure 17 and Figure 18 illustrate the IP address verification.

### 9.7. Analysis 7–Hibernet.py

Attack script recognized as Hibernet.py employed the python3 Hibernet.py http://42.1.63.189 command to operate. The attack script contained 70792 invalid HTTP requests generated through various public proxies to launch an attack over a web server. To successfully execute the attack script, a list of public proxy IP addresses are required and must be located within the proxy file. Initial inquiry indicated that necessary request headers were supplied; however, the existence of proxy headers was inconsistent. Proxy headers attached in the HTTP requests also vary. Firstly, only single proxy header existed such as X-Forwarded-For in one HTTP request. Secondly, double proxy headers known like X-Forwarded-For and VIA, thirdly triple proxy headers such as X-Proxy-ID with X-Forwarded-For and VIA. These patterns show that each proxy provider has different approaches to introduce the source connection from which the proxy originates from when establishing a web server. The various proxy headers utilized by the attack scripts are illustrated in Figure 19, Figure 20 and Figure 21.

Results obtained in this study also reveal that some of the HTTP requests were missing. The missing header is known as a referrer which commonly exists in an HTTP request. To ascertain that the headers were missing due to fake HTTP requests, the proxy IP address utilized in the attack script was used to launch a genuine HTTP request. This is to ensure that the missing header was not due to the proxy intentionally hidden the header. The proxy IP address was 36.66.55.181 and utilized port 8080 for clients to connect to the proxy. As a result, a genuine request against a web server using the proxy supplied all the common headers. Figure 22 indicates the header provided by the attack script while Figure 23 illustrates the header that adopted the proxy.

### 9.8. Analysis 8–UFONet.py

The final attack script executed in this study is the UFONet.py. The attack script generated minimal fake GET requests compared to the previous attack scripts that were executed. The attack script generates 51 HTTP requests which were launched through a proxy to attack a web server. The attack script was operated using the /ufonet -a http://42.1.63.189 -r 10 -threads 500 command. The preliminary analysis revealed that the attack script adopted a different name, signifying that an HTTP request came from a proxy by using the X-Pingback-Forwarded-For header. Aside from that, the command proxy publicly known as VIA was not present. Detailed analysis disclosed that the attack script generated an odd value for a user agent. User agent conveys client information including the browser name, operating system version, etc. Apart from that, an inconsistent referral header was detected and appeared in each HTTP request. The analysis also found that the attack script generated request queries that were attached together with the referral header along with the URL, thus indicating that the query was generated from the previous page and was forwarded to next page. Figure 24 demonstrates the GET header and the HTTP referral followed by Figure 25 which indicates the list of the user-agent.

## 10. Overview of Critical Analysis

This section elaborates the similarities and differences of the forged request headers constituted by the attack scripts executed in the previous section (Section 9.1 until Section 9.8). The most manipulated request headers will be analyzed critically including the user-agent, query string, referral, connection, accept-language and proxy headers. Although there are numerous request headers available that are involved during a genuine HTTP request transaction, this study advocates that the selection of the header as noted above were suffient to reveal the use of forged request headers when executing HTTP DDoS attacks.

### 10.1. GET Header-User-Agent

All attack scripts that have been executed contain user-agents. The user-agents delivered in all attack scripts are most likely to be equal as those found in a genuine GET header. However, there are several attack scripts that indicate a different appearance such as the HOIC attack script that supplied a minimal header with the absence of a user-agent. Apart from that, the Chihulk.py attack script adopted a user-agent that comes from a variety of platforms as it contains mobile device name, web crawler from Googlebot and video game devices such as Play Station. In contrast, The UFONet.py attack script delivers an odd user-agent as it comprises the URL and IP address which do not provide information about the requestor. The user agent is expected to supply information about the client including the web browser version, operating system, etc. Zhang, et al. [46] note that information contained in a common web browser. However, this study gathers contrasting findings in which none of the web browsers contain such information as explicated by the results derived from the UFONet.py attack script. Apart from that, Gou, et al. [47] noted that a genuine HTTP request contains headers such as “host”, “connection”, “accept”, “user-agent” or any headers that relate to the request. The user-agent is utilized by the web user’s browser by indicating the information of a request [29]. Figure 26 illustrates the user-agent that is included in the web browser.

### 10.2. GET Header–Request Query

The request query supplied by each attack script was contradictory to a genuine request as the attacker tried to mimic the request made by humans. The request query included in the Attack.py attack script was created using capital letters while the Chihulk.py attack script comprised of a number and a combination of few symbols. Emulating a legitimate request does not necessarily supply a request query as proven in the HOIC and Hibetnet.py attack scripts. The Golden Eye.py and BlackHorizon.py attack scripts adopted longer request queries constituting a combination of small letters, capital letters, numbers and symbols. Besides that, there is no predefined limit for HTTP request to process request headers and its values. Furthermore, an appropriate length for request header is difficult to be defined [48]. Request queries produced by the Wreckuests.py attack script delivered a brilliant strategy in mimicking a request query as it contains a query that can be understood by humans although it is irrelevant for a web server to process. Besides that, queries found in other attack scripts are not readable. Request query generated by the UFONet.py attack script contains query that is identical to Golden Eye.py and BlackHorizon.py attack scripts. However, the query is found to be shorter. It should be noted that the web server adopted in this analysis was designed with a static HTML page and was not designed to process a query. Hence, the continuous request query received has proven that the GET request is malicious.

### 10.3. GET Header-Referral

Almost all attack scripts supplied a referral header to show that the current web pages were accessible from the previous web page. According to Reid [49], a header referrer refers to the earlier address for web page browsed by users. In this context, any web addresses that belong to any website are permitted to be a referral as the existence of the referral address in terms of its relevance was not examined. This analysis has been successful at disclosing that all attack scripts have attached a referral with a valid URL address with the exception of the HOIC attack script. However, the source of the address is irrelevant to be a value for referral header for the current page that is being accessed. This pattern shows that the HTTP referral has been manipulated. Fielding, et al. [50] suggest that for HTTP version 1.1, a user has an option to select whether the referral header is sent as the referral which may contain a private link and the presence of a referral in an HTTP request may reveal confidential information. In terms of protocol showed in the referral header, all attack scripts utilize an HTTP protocol except for Blackhorizon.py attack script as it adopted a referral from an HTTPS protocol that contained Facebook address, which provides an understanding that a web page is accessible from the previous web page which is from Facebook.

Most of the attack scripts generated referrals that come from a search engine to show that the current web page was previously searched and is accessible from a search engine which are commonly performed by genuine users. On the other hand, the referral is supposed to be encrypted as it has the ability to reveal the web browsing history and information that belonged to the user that may exist in these headers [51]. Dolnak [52] addresses the problem highlighted by Fielding and Reschke [51] where they suggested a referrer header policy that can control information indicated by the referrer to reduce the risk of information leakage. However, the proposed solution does not curb HTTP referrer manipulation created by HTTP DDoS attacks.

### 10.4. GET Header-Connection and Accept Language

HTTP requests contain standard value and header which have been defined by the International body. However, the existence of incorrect headers and value signal that the GET header has been manipulated. During the execution of Attack.py and Chihulk.py attack scripts, the connection header marks a close which entails that a web server has stopped receiving GET requests. However, in the case on these attack scripts, the header marks a close with the web server receiving repetitive HTTP requests. Besides, the Keep-Alive is supposed to be the header for an HTTP response and the value for a connection header [53]. The appearance of Keep-Alive in the request header produced by *Golden* Eye.py and BlackHorizon.py attack scripts indicates that attackers accidentally use the header to emulate a user’s request when executing HTTP DDoS attacks. In addition, the Wreckuests.py, Hibernet.py and UFONet.py attack scripts do not attach Keep-Alive as the header while the HOIC attack script contains a minimal GET header with the absence of a connection header.

The Accept-Language in a request header indicates the web browser language used by users and the existence of this header in an HTTP request can be used to identify the geolocation of users [49]. This study found that the Accept-Language header was not generated by five attack scripts (Attack.py, Chihulk.py, GoldenEye.py, BlackHorizon.py and Wreckuests.py). Three attack scripts (GoldenEye.py, Hibernet.py and UFONet.py) indicate an inconsistent appearance as the header was missing for certain HTTP requests. The Accept-Language shows the language utilized by users in web browser and the missing of this header indicate that the source initiators of an HTTP request was generated from automated tools. Clients adopt web browsers to browse through the web server content, which will create an HTTP request. The Connection and Accept-Language headers are the general headers that exist in an HTTP request [47].

### 10.5. Proxy Headers and IP Spoofing

The HTTP request will provide proxy headers if the source requestor adopted a proxy. The Hibernet.py and UFONet.py attack scripts comprise of many proxies, indicating that proxy headers were not displayed by certain proxy providers. Another method utilized by the attacker was IP spoofing to execute HTTP DDoS as adopted by the Wreckuests.py attack script. The usage of IP address spoofing to launch the attack poses difficulty as the attacker is capable of attaching the proxy header thereby making the request to appear as if it comes from proxy.

The Hibernet.py attack script generated two IP addresses in X-Forwarded-For headers in which the second IP address belongs to this study’s equipment and the first IP is believed to originate from another proxy. Petersson and Nilsson [54] noted that a proxy require to establish a connection to another proxy when it is necessary to acquire the web server content and utilize headers known as X-Forwarded-For. The UFONet.py attack script delivers a proxy header named as X-Pingback-Forwarded-For which does not equal to the standard proxy header known as X-Forwarded-For. The findings support the explanation made by Petersson and Nilsson [54]. In this study, the existence of distinct proxy headers and the inconsistent appearances due to proxy header are not updated by proxy provider and that the headers are optional to be displayed in the GET request header.

## 11. Results

Based on the execution of eight attack scripts, this analysis has successfully discloses seven (7) forged request headers patterns produced by HTTP DDoS. The request header pattern for each attack script executed in this study has been thoroughly discussed in Section 9.1 to Section 9.8. Results of the study are further supported by a critical analysis reported in the previous section (Section 10.1 to Section 10.5) which provide a strong justification to assert that HTTP DDoS has a specific pattern to generate forged HTTP requests. This study’s findings corroborated with the findings obtained in prior studies with regard to the request header patterns, which pave the way for future studies to apply appropriate approach when dealing HTTP DDoS attack. HTTP DDoS attacks are capable of generating a substantial request in a minimal duration as the source of the request originates from various platforms. The use of attack scripts in this study comprises HTTP DDoS from a direct attack, both through proxy and spoof IP address. Hence, based on the attack scripts utilized in this study, the architecture of HTTP DDoS has been successfully designed. Additionally, the taxonomy of the forged request headers has also been designed including false request headers mentioned in prior studies described in Section 5. The architecture of HTTP DDoS attack is illustrated in Figure 27 followed by descriptions regarding forged GET headers produced by HTTP DDoS attacks. The taxonomy of the forged request headers is depicted in Figure 28.

**Automated Tools:** HTTP DDoS attack requires enormous HTTP requests to overwhelm web servers. Hence, attackers are required to adopt automated tools to generate massive requests which only require a minimal duration to create forged request traffics. A massive GET requests received results in the web server to become unresponsive due to overloading as it reaches beyond its capacity to process HTTP request.**Incomplete Request Headers:** HTTP DDoS attacks generate an incomplete request header as it is unnecessary to supply complete headers as the purpose of the attack is to gain a server’s resources to cause the web server to be unresponsive. Attackers are able to generate substantial HTTP requests without a request header. However, to create complexities in detecting the attack, attackers must supply the request headers to emulate user’s access pattern to avoid from being detected.**Incorrect Connection Status:** The connection status header in an HTTP request presents the status of the client which has an active connection with a web server. However, during an occurrence of HTTP DDoS attack, the status is marked as close with continuous HTTP requests received. Apart from that, the existence of Keep-Alive in the request header is a sign of HTTP DDoS as it is not the header for an HTTP request.**Irrelevant Request Header Component:** The existence of request headers in an HTTP request has been explained in the request for comments (RFC) which is hosted by the Internet Engineering Task Force (IETF). Hence, the appearance of non-standard request headers in a repeated HTTP request provides a strong indicator that the source of the request originates from an attacker.**Irrelevant Request Query and Query Length:** A client will search for information that relates to the web server’s content and adopts a human language. This search will then be translated into a request query to forward the search to the web server to be processed. An HTTP DDoS attack emulates this process by creating a request query which contains either special characters such as ‘!’, ‘@’, ‘#’, ‘$’, and ‘^’ or a longer query.**Irrelevant Value for HTTP Referrer:** The value for HTTP referrer is the previous URL accessed by a user. An HTTP DDoS attack is capable of mimicking this operation by attaching a valid URL. Although the attack is able to supply the valid URL, it is irrelevant to be the referrer of the current web page or website. A genuine request will indicate that the source of the referral must come from valid resources.**False Proxy Header:** The existence of a public proxy that can be utilized freely causes the service to be misused by attackers to execute HTTP DDoS attacks. Besides, there is no restriction to adopt the public proxy which further increases attackers’ capability to launch attacks without being detected. HTTP DDoS attacks launched through a public proxy deliver inconsistent proxy headers that pose difficulties in recognizing whether the source of a request originates from proxy or directly from the client as well as in detecting whether the source request was legitimate or malicious.**Spoof IP Address:** An attacker utilizes a spoof IP address to execute HTTP DDoS attack to conceal the activity of overwhelming a web server. The analysis conducted in Section 9.6 indicates that the Wreckuests.py attack script adopted several machines and used public IP addresses with an attachment of proxy headers to make the source request to appear as if they originate from a proxy. The execution of this attack script provides a significant finding with regards to HTTP DDoS attack strategy discussed earlier in Section 4.

## 12. Attack Category Associated with Real HTTP DDoS

The attack script adopted in this research is well-known and has been utilized by many prior studies. However, none of the past studies perform any mapping of the attack scripts in terms of showing the linkage between the type of the attack script and its attack strategy. Thus, based on the attack patterns executed, this study suggests that the attack scripts belong to five categories: web proxies, constant, server load, main page attack and random attack. Table 9 presents details regarding the dataset mapping with its attack strategy.

## 13. Conclusions

In this paper, an analysis of forged request headers created by HTTP DDoS was conducted. It is important to note that the request header is not compulsory to be presented which enables cyber intruders to generate forged headers that are identical to legitimate HTTP request. This research adopted eight attack scripts which have been utilized and executed separately to obtain precise information regarding the attack patterns. The attacks were executed in both the internet and external networks to observe the attack behavior in launching the attacks in these networks. A critical analysis was conducted, and the results show that there are similarities and differences between false request headers generated by various attack scripts in both internal and external networks. The results thus indicate that manipulation of request headers was constituted in various ways by HTTP DDoS to make the HTTP request appear authentic. Apart from that, the request headers require a major improvement particularly in the area of security to prevent the header from being manipulated by attackers. The reinforcement of the security for an HTTP protocol will ensure that HTTP DDoS attacks can be easily detected in future.

## 14. Future Work

Future research is recommended to include an execution of HTTP DDoS attacks in HTTP version 2 to observe the attack strategy in the manipulation of request headers. HTTP version 2 has a different structure compared to HTTP version 1.1. Hence, further exploration investigating an execution of HTTP DDoS in HTTP version 2 to analyze the request header structure which can possibly be forged by HTTP DDoS attacks. As technology evolves, the approach for a web browser communicating with a web server also changes. Currently, most of the web browsers utilize HTTP version 1.1 whereby 90% of web servers are not migrating to HTTP version 2 [55] as the migration is time consuming. The structures of HTTP version 2 still support basic features existed in HTTP 1.1 [56]. [57] also noted that the HTTP version 2 still employs several request header components which are used by HTTP version 1.1 such as user-agent, accept and accept-language as highlighted in Table 10.

Ludin and Garza [57] explained that only unique bytes were sent to a web server unlike HTTP 1.1 whereby the entire bytes are delivered to a web server. Table 10 illustrates the explanation pertaining to unique byte where the second request only sends the unique byte, which is only 10 bytes.

## 15. Possible Solutions and Open Issues

The outcome of this study indicated in Section 9 successfully reveals eight forged request headers constituted by HTTP DDoS attacks which leads to possible solutions for future researchers to further explore the various ways DDoS attacks are executed at the application layer. Among the possible areas for future research include:(a)Development of a web browser without adopting secure socket layers to encrypt the GET header and its value to prevent header manipulation.(b)Analysis of other devices acted as botnets which come from IoT devices to observe the forged GET header patterns.(c)Execution of HTTP DDoS through HTTPS protocol and HTTP version 2 to explore the possible GET header manipulation.(d)Development of new standards to reject HTTP request originating from a non-browser as done by HTTP DDoS adopted attack scripts to generate substantial requests.

Future researchers should also take note that vulnerabilities at the application layer not only lead to the occurrence of several types of attacks, but the shortfalls can also be utilized to design a defense system as discussed in Section 5. Future research should also take precautionary measures as there may be possible open issues that require further attention when providing solutions in detecting HTTP DDoS, including:(a)The appearance of HTTP DDoS as the service increases the occurrence of the attack and becomes simpler to be executed.(b)The existence of public proxies scattered around the world that provides opportunity for cyber intruders to utilize the service to launch HTTP DDoS attacks. The usage of proxies to execute HTTP DDoS increases the complexity to trace the source of the request. Besides that, the usage of proxies does not require any authentication which allows both genuine users and attackers to adopt the same proxy.(c)A web server accepts any requestor without performing an inspection of the platform’s source such as whether it comes through scripting, tools or web browsers. A web server is supposed to be more selective when accepting eligible clients to be connected to a web server to gain access to the web server’s content.(d)The structure of the HTTP protocol allows the request headers and its data to be edited which leads to manipulation of request headers. A restriction should be applied to allow only specific request headers to prevent any header manipulation executed by attackers.(e)The request header presented in HTTP request transaction should take note of the requestor information such as the web browser’s name, language, operating system, etc. However, these headers are not compulsory to appear in each HTTP request transaction. Due to this circumstance, attackers are only required to generate an HTTP request without having any request header to overwhelm a web server. Besides, attackers often adopt request headers to make the header to appear authentic which lead to further complication in detecting both the DDoS and the authenticity of the request at the application layer.(f)The existence of attack tools and scripts available on the Internet allows an HTTP DDoS to be easily executed. The appearance to the attack script provides an opportunity for researchers in conducting further research. However, the same attack script could also be adopted by an attacker as a medium to launch an attack.(g)The proxy provider adopted a distinct name to indicate that a HTTP request originates from a proxy. The usage of a different name for a proxy header leads to confusion. Besides, the same proxy is likely to be adopted by both an authentic client and an attacker to access the same web server.(h)A high-speed Internet delivers an advantage to users to browse online contents. However, the Internet speed provides a significant impact to substantial amount of HTTP requests that can be generated through a single requestor within a specific time period.(i)The attack scripts adopted in this research were publicly available and are compulsory to be executed in the real network. A researcher requires a sound technical skill to configure those devices to simulate the attack in the Local Area Network (LAN) or Wide Area Network (WAN). The impact of an attack executed in the WAN network requires a higher attention as the Internet Service Provider (ISP) has possibly blocked the traffic therefore jeopardizing the research results.(j)The development of a defense system which utilizes either signature-based, anomaly-based and hybrid detection methods must be selected with precaution as each detection method has its own strengths and limitations.

## Figures and Tables

**Figure 1 sensors-20-03820-f001:**
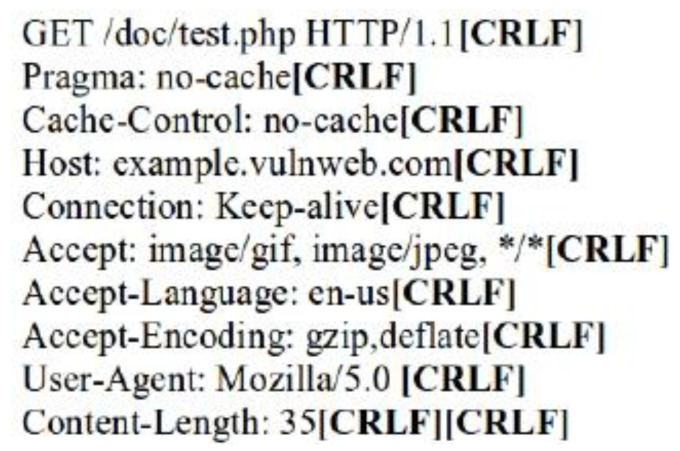
Two CRLF in last request header (Genuine).

**Figure 2 sensors-20-03820-f002:**
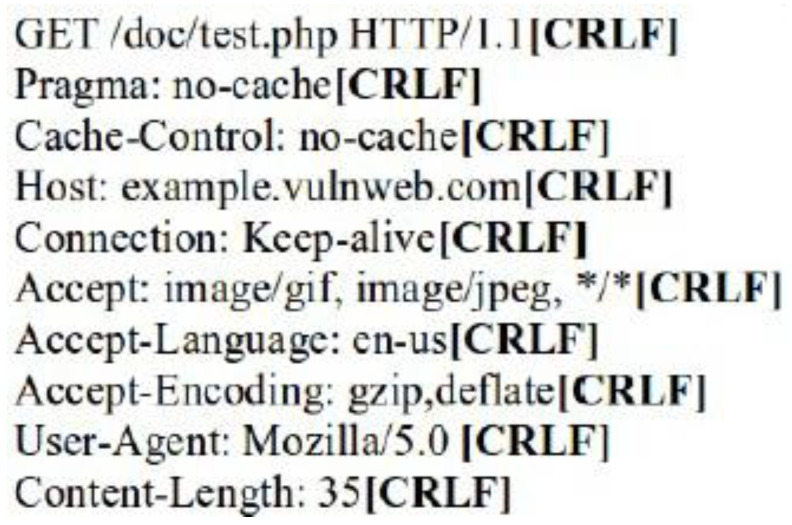
Single CRLF in last request header (Forged).

**Figure 3 sensors-20-03820-f003:**
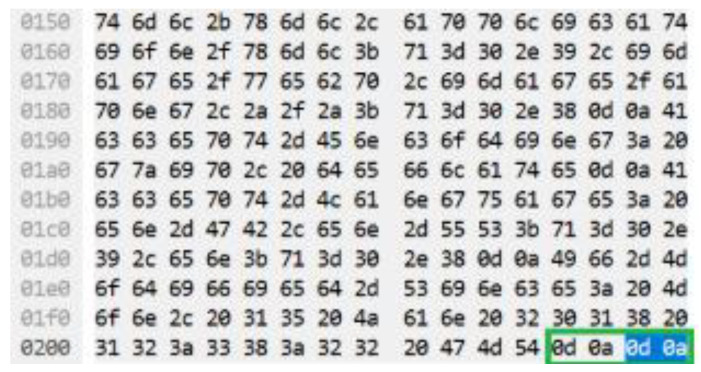
ASCII code for valid HTTP request.

**Figure 4 sensors-20-03820-f004:**
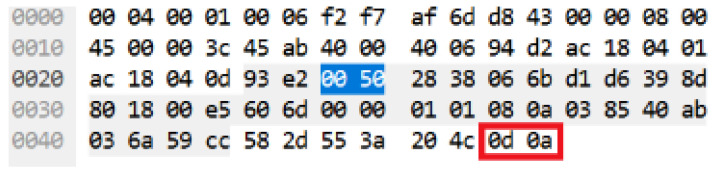
ASCII code for attack HTTP request

**Figure 5 sensors-20-03820-f005:**
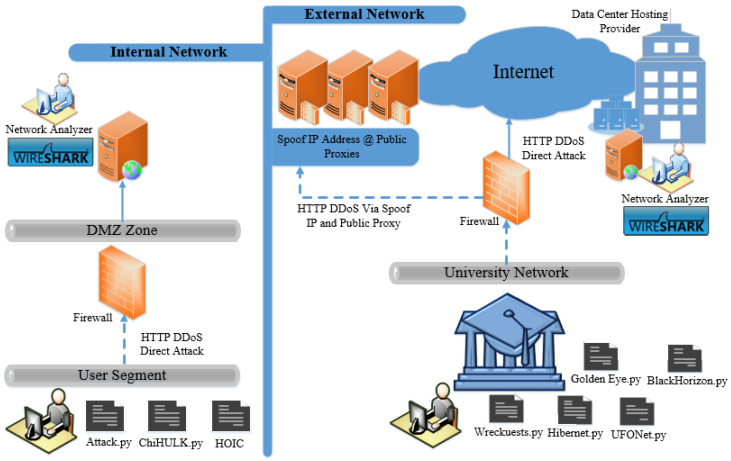
Attack script executed in internal and external networks.

**Figure 6 sensors-20-03820-f006:**
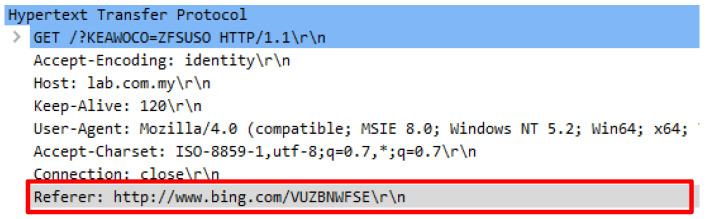
Forged referral from www.big.com search engine.

**Figure 7 sensors-20-03820-f007:**
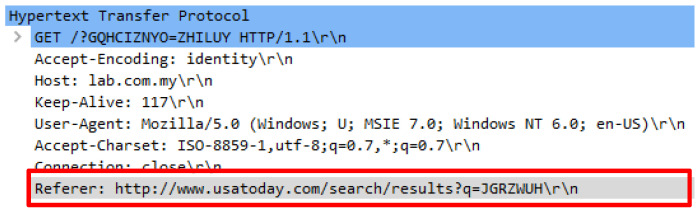
Forged referral from www.usatoday.com new web site.

**Figure 8 sensors-20-03820-f008:**
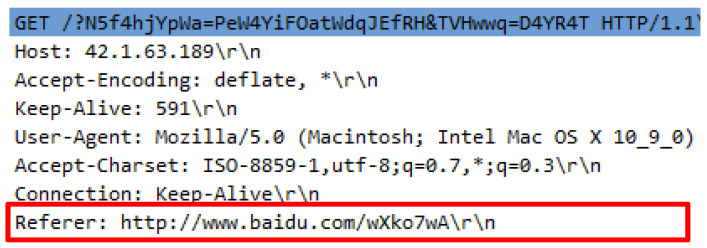
Request headers with referrer.

**Figure 9 sensors-20-03820-f009:**
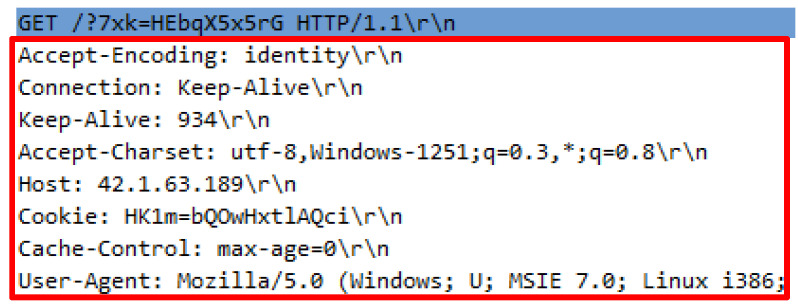
Request headers with missing referrer.

**Figure 10 sensors-20-03820-f010:**
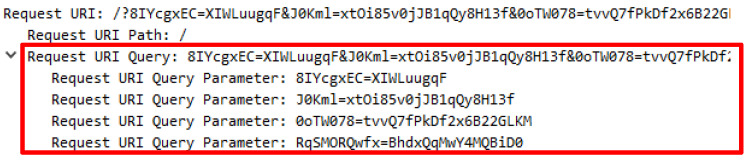
False request query in DTS3.

**Figure 11 sensors-20-03820-f011:**
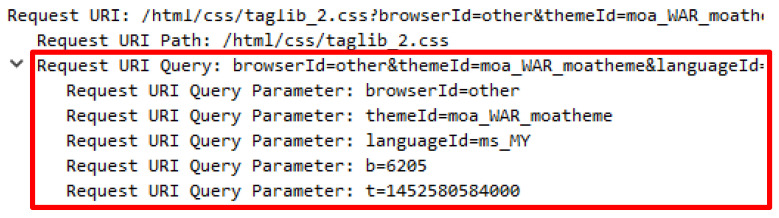
Genuine request query generates by server.

**Figure 12 sensors-20-03820-f012:**
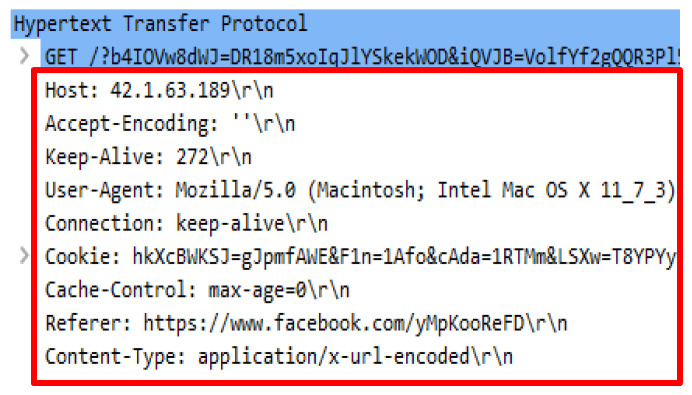
DTS5 request headers.

**Figure 13 sensors-20-03820-f013:**
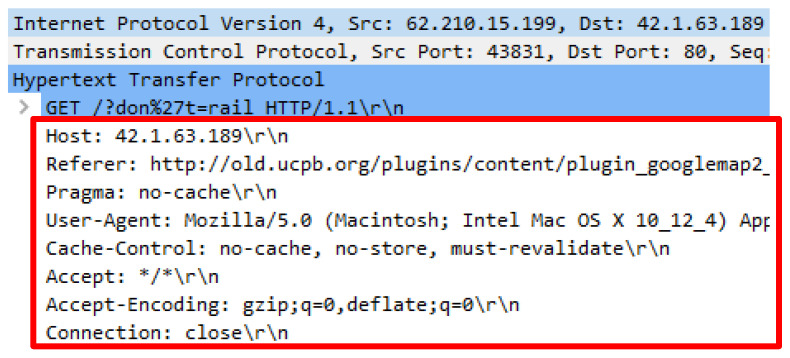
Request headers generate by IP address 62.210.15.199.

**Figure 14 sensors-20-03820-f014:**
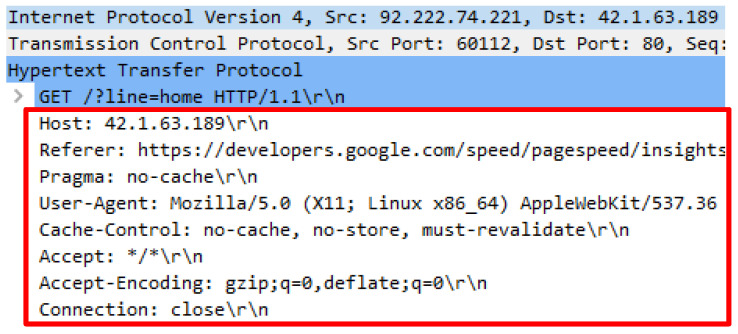
Request headers generate by IP address 92.222.74.221.

**Figure 15 sensors-20-03820-f015:**
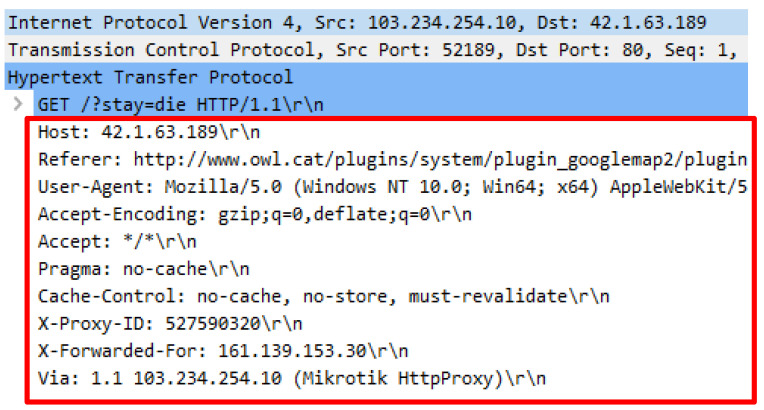
Request headers generate by IP address 103.234.254.10.

**Figure 16 sensors-20-03820-f016:**
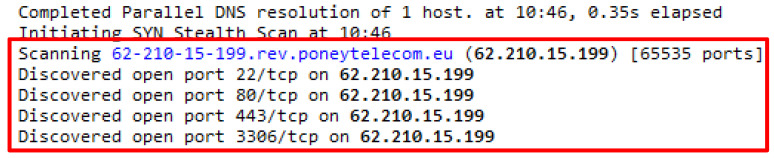
IP address source verification for 62.210.15.199.

**Figure 17 sensors-20-03820-f017:**
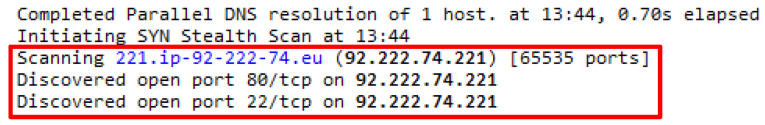
IP address source verification for 92.222.74.221.

**Figure 18 sensors-20-03820-f018:**
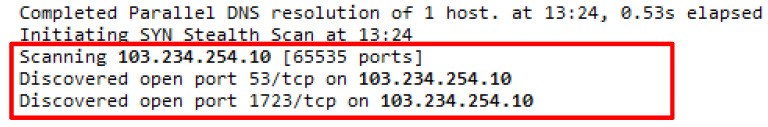
IP address source verification for 103.234.254.10.

**Figure 19 sensors-20-03820-f019:**
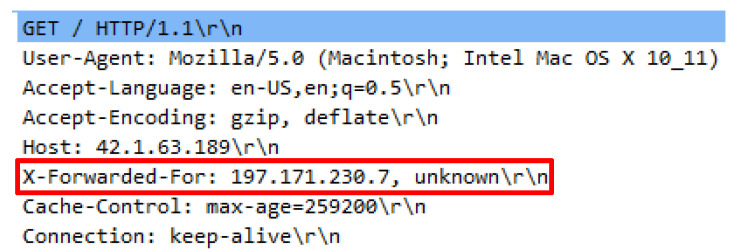
X-Forward-For in request header.

**Figure 20 sensors-20-03820-f020:**
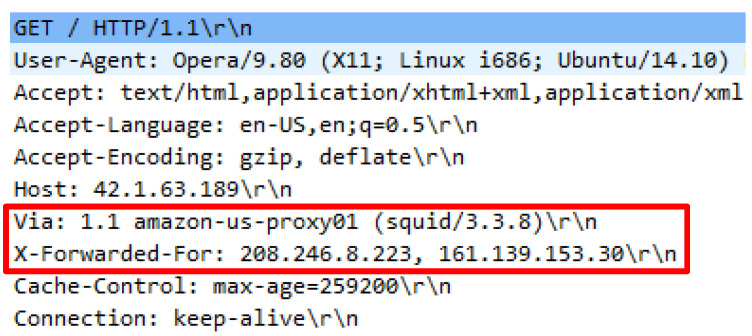
X-Forward-For and Via in request header.

**Figure 21 sensors-20-03820-f021:**
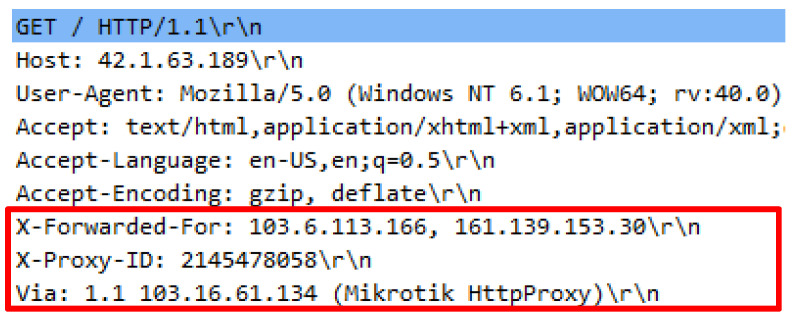
X-Forward-For, Via and X-Proxy-ID in request header.

**Figure 22 sensors-20-03820-f022:**
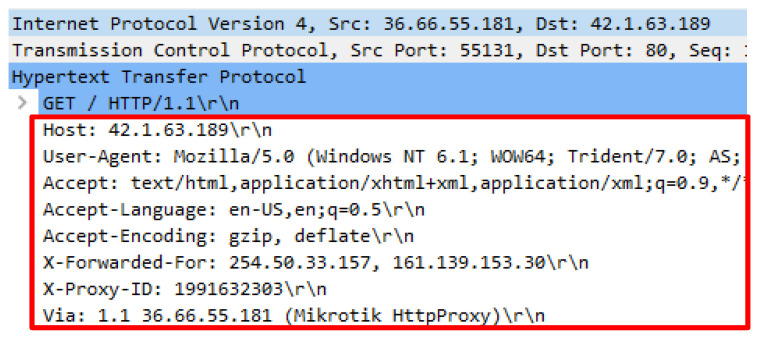
Missing referrer in HTTP request.

**Figure 23 sensors-20-03820-f023:**
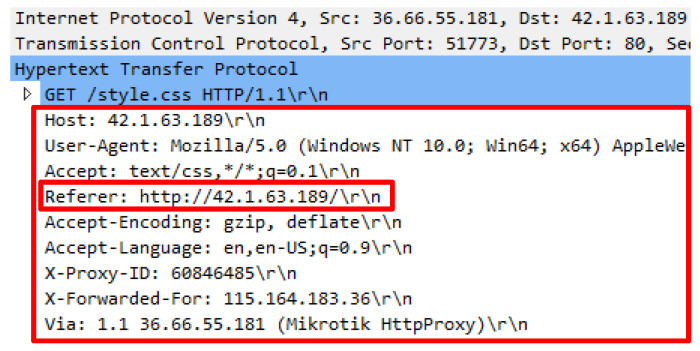
Complete HTTP request with referrer.

**Figure 24 sensors-20-03820-f024:**
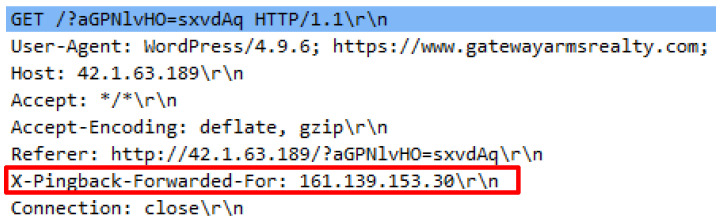
Proxy header in DTS8.

**Figure 25 sensors-20-03820-f025:**
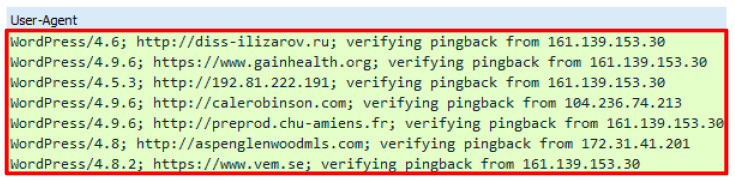
User agent in DTS8.

**Figure 26 sensors-20-03820-f026:**
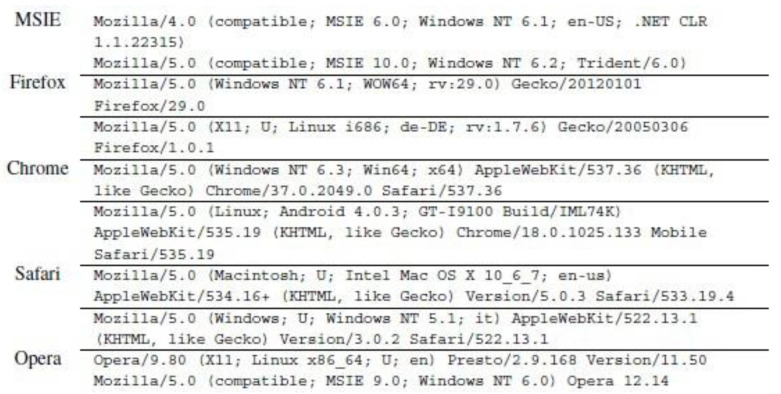
User-Agent in Web Browser.

**Figure 27 sensors-20-03820-f027:**
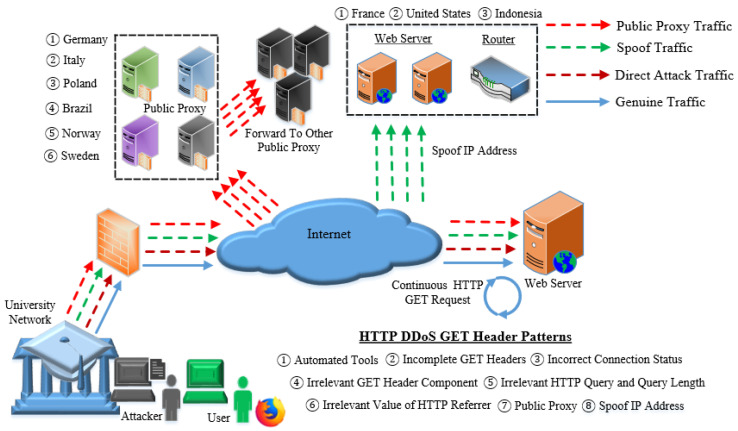
Architecture of HTTP DDoS attack.

**Figure 28 sensors-20-03820-f028:**
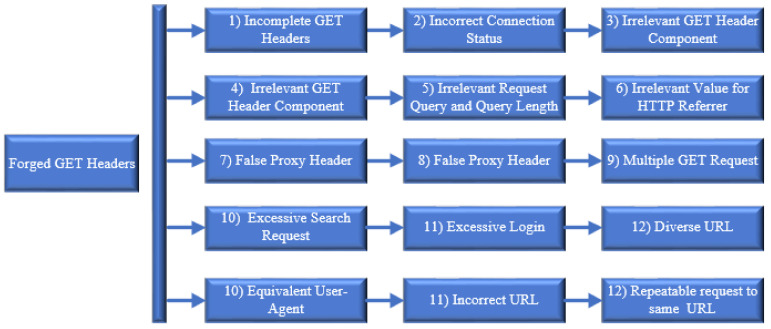
Taxonomy of Forged GET headers by HTTP DDoS attack.

**Table 1 sensors-20-03820-t001:** Hardware and software for analysis.

Hardware	Software	Software Function
Web Server Processor Intel(R) Core (TM) i7-6700 CPU @ 3.40GHz 8GB Memory	Python coding editor and Compiler	Edit and execute attack scripts
	Notepad++	View and edit attack scripts
	Proxy Switcher	Identify active public proxies
Client Processor Intel(R) Core (TM) i7-3770 CPU @ 3.40GHz 12GB Memory	Nmap	Identify port usage
	Wireshark	Analyze HTTP request traffics
Firewall	Attack Script	HTTP DDoS Code
	Open Source Operating System: Ubuntu	Provide platform to execute HTTP DDoS attack
	Windows Server 2016 Windows 8 Pro	To host web-based application and initiate HTTP request

**Table 2 sensors-20-03820-t002:** Analysis attribute.

No.	Script Name	Attack Scale	Attack Pattern	Attack Duration	Target URL
1.	Attack.py	Internal Attack	High rate direct attack	10 Min	http://lab.com.my
2.	Chihulk.py		High rate direct attack	10 Min Internal	http://lab.com.my
3.	HOIC				
4.	Golden Eye.py	External Attack	High rate direct attack	5 Min	http://42.1.63.189
5.	BlackHorizon.py				
6.	Wreckuests.py	External Attack	High rate through proxy	5 Min	http://42.1.63.189
7.	Hibernet.py				
8.	UFONet.py				

**Table 3 sensors-20-03820-t003:** User Agent and Request Query.

No.	User Agent String	Request Query
1.	Mozilla/4.0 (compatible; MSIE 8.0; Windows NT 5.2; Win64; x64; Trident/4.0)	KEAWOCO = ZFSUSO
2.	Mozilla/5.0 (Windows; U; MSIE 7.0; Windows NT 6.0; en-US)	QJCQABP=MIGMQXRML
3.	Mozilla/5.0 (Windows; U; Windows NT 5.2; en-US; rv:1.9.1.3) Gecko/20090824 Firefox/3.5.3 (.NET CLR 3.5.30729)	QJCQABP=MIGMQXRML
4.	Mozilla/4.0 (compatible; MSIE 6.1; Windows XP)	DYH=GFOUW
5.	Opera/9.80 (Windows NT 5.2; U; ru) Presto/2.5.22 Version/10.51	GQHCIZNYO=ZHILUY
6.	Mozilla/5.0 (Windows; U; MSIE 7.0; Windows NT 6.0; en-US)	DYH=GFOUW

**Table 4 sensors-20-03820-t004:** Fake user agent.

No.	User Agent String	Request Query
1.	Mozilla/5.0 (iPad; U; CPU OS 3_2 like Mac OS X; en-us) AppleWebKit/531.21.10 (KHTML, like Gecko) Version/4.0.4 Mobile/7B334b Safari/531.21.10	\357\277\275\357\277\275{\357\277\275\177 =\357\277\275\357\277\275y
2.	BlackBerry8300/4.2.2 Profile/MIDP-2.0 Configuration/CLDC-1.1 VendorID/107 UP.Link/6.2.3.15.0	\357\277\275\357\277\275\357\277\275 =\357\277\275\357\277\275\357\277\275
3.	BlackBerry9000/5.0.0.93 Profile/MIDP-2.0 Configuration/CLDC-1.1 VendorID/179	y\357\277\275\357\277\275\357\277\275\357\277\275\357\277\275 {\357\277\275=\357\277\275\357\277\275\357\277\275
4.	Mozilla/5.0 (compatible; bingbot/2.0; +http://www.bing.com/bingbot.htm)	~\357\277\275\357\277\275z}\357\277\275 {~=\357\277\275\357\277\275\357\277\275\357\277\275
5	Googlebot/2.1 (http://www.googlebot.com/bot.html)	}\357\277\275\357\277\275=\357\277\275\357\277 \275z\357\277\275\357\277\275~\357\277\275
6.	Mozilla/5.0 (PLAYSTATION 3; 2.00)	z\357\277\275\357\277\275\357\277\275y=\357\277\275

**Table 5 sensors-20-03820-t005:** HTTP referrer with request query.

No	HTTP Referral with Request Query
1.	http://filehippo.com/search?q=\221y\203\231\213{\214\217\222\215\r\n
2.	http://taginfo.openstreetmap.org/search?q=~\177\235z\227\236\r\n
3.	http://www.baoxaydung.com.vn/news/vn/search&q=\225\211\224\235|\240\227\215\216\r\n
4.	https://steamcommunity.com/market/search?q=\217x\205\203\226\235{\r\n
5.	https://www.npmjs.com/search?q=\212\205\207x}\220\232\217\217\r\n

**Table 6 sensors-20-03820-t006:** Valid and invalid HTTP referrer with request query.

Authentic	Fake
http://ytmnd.com/search?q=catch+that+man\r\n	http://ytmnd.com/search?q=\215\231|\211~\216\230\r\n
http://millercenter.org/search?search=president\r\n	http://millercenter.org/search?q=\214{{\205\240\220\211\r\n

**Table 7 sensors-20-03820-t007:** Complete and incomplete request headers.

Complete GET Header (Genuine)	Incomplete GET Header (HTTP DDoS)
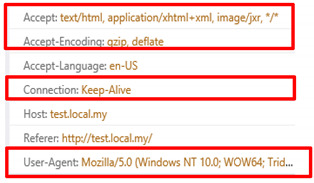	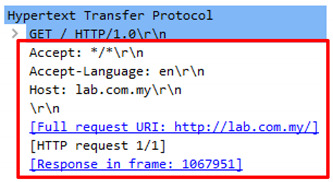

**Table 8 sensors-20-03820-t008:** Valid and invalid Keep-Alive values.

Valid Keep Alive from HTTP Response	Invalid Keep-Alive from GET Header
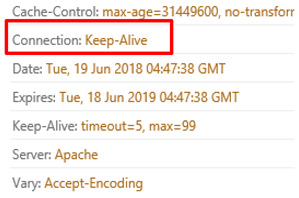	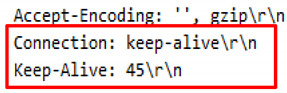

**Table 9 sensors-20-03820-t009:** Attack script mapping with existing attack strategy.

No	Attack Strategy	Attack Script Name
1.	Web Proxies	Hibernet.py
		UFONet.py
2.	Spoofing	Wreckuests.py
3.	Server Load	Chihulk, High Orbit Ion Canon (HOIC), Golden Eye, BlackHorizon
4.	Main Page Attack	
5.	Random Attack	

**Table 10 sensors-20-03820-t010:** HTTP 2 GET headers.

Request 1 # Total Bytes 220	Request 2 # Total Byte 230
: Authority: www.akamai.com : Method: GET : Path:/ : scheme: https accept: text/html,application/xhtml+xml accept-languange:en-US,en;q=0.8 **cookie:last_page=286A7F3DE** upgrade-insecure-request:1 user-agent:mozilla http2	: Authority: www.akamai.com : Method: GET : Path:/style.css : scheme: https accept: text/html,application/xhtml+xml accept-languange:en-US,en;q=0.8 **cookie:last_page=*398AB8E8F** upgrade-insecure-request:1 user-agent:mozilla http2

* Those in bold mark the difference.

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
