# Peer review of "Recent Analysis of Forged Request Headers Constituted by HTTP DDoS"

_sensors, 2020, doi:10.3390/s20143820_

Round 1

Reviewer 1 Report

  1. The first paragraph (From line no. 32 to 45) is not needed in your paper. The paper needs only necessary statements, not common sense.
  2. There are so many English grammar errors such as “reveal” in line no. 78 and line 86.
  3. There is no section 2.4 in your manuscript, which you state in line 260. Is it section 5 ?
  4. Where is section 3.1 and 3.2 in your manuscript ?
  5. Why did you select the designated 8 DDOS tools ? Are there more tools ? Describe the reason why you selected these tools.

Reviewer 2 Report

I found the paper interesting and the topic valuable.

To consider the paper for publication a number of issues MUST be solved:

1) English MUST be improved.

2) The paper deals with DDOS but in the experimental phase the Distributed part is not emphatized. The experimental part should consider distributed experimental platforms like Planetlab or Bismark. At least at qualitative level.

3) Header manipulation (like HTTP header manipulation) is a common practice in the field of both security and censorship [ref3]. This should be discussed in the paper.

4) The paper falls in the important area of DDOS both anomaly and signature based. The introduction MUST be rewritten by discussing these aspects. A large number of important references are missing [ref4-10].

5) The authors should discuss the impact of HTTP2 on the proposed analysis (at least qualitatively).

6) In the figures reporting the forged field should be useful to circle in red the forged field.

7) Sections 13 and 14 should be merged and revised in way that adher to a journal publication. The current version is too naive and unprofessional.

[ref1] https://www.planet-lab.org/

[ref2] https://www.measurementlab.net/tests/bismark/

[ref3] https://doi.org/10.1016/j.comnet.2015.03.008

[ref4] https://link.springer.com/article/10.1007/s00500-014-1250-8

[ref5] https://doi.org/10.3233/JCS-2009-0350

[ref6] https://www.hindawi.com/journals/jcnc/2017/7674594/abs/

[ref7] https://ieeexplore.ieee.org/abstract/document/7155851/

[ref8] https://dl.acm.org/doi/abs/10.1145/997150.997156

[ref9] https://doi.org/10.1109/SURV.2013.031413.00127

[ref10] https://doi.org/10.1016/j.comnet.2003.10.003

Round 2

Reviewer 1 Report

  1. First, sort the references by alphabetical order of the first author and put the serial number for the reference. Use the number for citing in the text of your paper.
  2. The method to cite a reference is not correct in your paper. Change the method for all citing reference like “Jazi, Gonzalez, Stakhanova, and Ghorbani (2007) to “Jazi et. Al[#no]”.

Reviewer 2 Report

Authors have considered (almost) all the required changes.

Author Response

Dear Reviewer,

Thank you very much for reviewing our manuscript for second times. We have carried out minor changes as required by reviewer. The responses from the author are shown as follows:

Point 1: Moderate English changes required 

Response 1: Thank you for pointing this out. Changes has been made in line number 44 until 46 to make the explanation more precise. Add the sentence in line 60- 65 as improvement of the introduction.

We hope that you find our responses satisfactory and the manuscript is now acceptable for publication. Thank you.